# Recruiting and Engaging American Indian and Alaska Native Teens and Young Adults in a SMS Help-Seeking Intervention: Lessons Learned from the BRAVE Study

**DOI:** 10.3390/ijerph17249437

**Published:** 2020-12-16

**Authors:** David Stephens, Roger Peterson, Michelle Singer, Jacqueline Johnson, Stephanie Craig Rushing, Allyson Kelley

**Affiliations:** 1Northwest Portland Area Indian Health Board, 2121 SW Broadway #300, Portland, OR 97201, USA; dstephens@npaihb.org (D.S.); rpeterson@npaihb.org (R.P.); msinger@npaihb.org (M.S.); jackiejohnson30@gmail.com (J.J.); 2Allyson Kelley & Associates, Principal, 69705 Lake Drive, Sisters, OR 97759, USA; kelleyallyson@gmail.com

**Keywords:** American Indian, Alaska Native (AIAN), adolescent, recruitment and retention, help-seeking skills, mHealth, SMS intervention

## Abstract

This paper shares lessons learned recruiting and engaging participants in the BRAVE study, a randomized controlled trial carried out by the Northwest Portland Area Indian Health Board and the mHealth Impact Lab. The team recruited 2330 American Indian/Alaska Native (AI/AN) teens and young adults nationwide (15–24 years old) via social media channels and text message and enrolled 1030 to participate in the 9 month study. Teens and young adults who enrolled in this study received either: 8 weeks of BRAVE text messages designed to improve mental health, help-seeking skills, and promote cultural pride and resilience; or 8 weeks of Science Technology Engineering and Math (STEM) text messages, designed to elevate and re-affirm Native voices in science, technology, engineering, math and medicine; and then received the other set of messages. Results indicate that social media channels like Facebook and Instagram can be used to recruit AI/AN teens and young adults. Retention in this study was high, with 87% of participants completing both the BRAVE and STEM intervention arms. Lessons learned from this process may help teen and young adult-serving organizations, prevention programs, policy makers, researchers, and educators as they support the next generation of AI/AN change makers.

## 1. Introduction

To be effective, health and wellness interventions must reflect the cultural values, social contexts, and health epistemologies of American Indian and Alaska Native (AI/AN or Native) teens and young adults. Culturally tailored interventions are critically needed to increase the degree to which health messages are perceived as personally relevant by AI/ANs, to inspire and support behavior change [1]. From 2015 to 2018, the Northwest Portland Area Indian Health Board (NPAIHB) carried out formative research to design a multimedia, behavioral intervention to address alcohol misuse, intimate partner violence, suicidality among AI/AN teens and young adults, by modeling healthy social norms and help-seeking skills. The BRAVE intervention is the first of its kind using culturally relevant images, narrative role model videos, and help-seeking resources delivered via text message.

Having strong help-seeking skills and access to and awareness of help-seeking resources may help teens and young adults navigate common risky situations involving family and friends, including drug and alcohol misuse, dating violence, and suicidality. Previous research with AI/AN teens and young adults indicates that they are more likely to seek help from informal sources than formal sources [2] and males are less likely to seek help than females [3]. Factors known to promote help-seeking include emotional competence, positive attitudes, and social influences. 

According to the Center for Native American Youth, suicide prevention and mental health promotion remain one of the most daunting challenges for AI/AN communities [4]. Among AI/AN youth in 9th to 12th grade, the past year prevalence of suicidal thoughts, planning, and attempts were nearly 15% in 2017 [5]. Suicide is the second-leading cause of death for Native youth 10–24 years old; a rate that is 2.5 times higher than the national average [6]. Multilevel interventions are critically needed to stem the devastating, reverberating impact that adolescent suicide has in native communities [7]. AI/AN young adults experience violence, substance misuse, aggression, and limited opportunities to develop help-seeking skills—Native young men experience these inequities at higher rates than females [3,8]. Technology-based health interventions are emerging as an effective strategy for promoting health and wellbeing in AI/AN populations.

### 1.1. mHealth Access in Indian Country 

While the speed and quality of internet access and cell phone coverage is highly variable in native communities across the U.S., it is swiftly and steadily improving [9]. In 2009, AI/AN youth and young adults (13–21 years old) in the Pacific Northwest reported frequent media technology use, mirroring patterns reported by teens in the general population [10]. Similarly, a survey conducted in 2016 with 675 AI/AN youth nationwide about their media technology use found that 78% reported regular access to a smartphone, and 46% had regular access to a computer [11]. Over 62% of youth in this survey reported getting health information from the internet weekly or monthly, and 66% reported getting health information from social networking sites as often [11].

Guided by this information, the Northwest Portland Area Indian Health Board (NPAIHB) launched We R Native in 2011—a multimedia health resource for AI/AN teens and young adults. Guided by the core principles of positive youth development, We R Native promotes cultural identity and pride; provides age-appropriate health information addressing physical, mental, social, and spiritual health; amplifies youth-friendly messages that reflect healthy social norms; promotes education, community engagement, and community service; and offers safe “spaces” where AI/AN youth can feel connected to other AI/AN youth [12].

In addition to the website (weRnative.org), online resources include an Ask Auntie/Ask Uncle Q&A service, a text message service (text NATIVE to 97779), a YouTube channel, and social media accounts (Facebook, Instagram, Twitter). In 2019, the We R Native website had over 260,000 unique users. On average, the site receives over 25,000 pageviews per month, with 70% visiting from a mobile device. We R Native’s 780+ YouTube videos have 5.2 million impressions, and its text messaging service has over 5800 subscribers. We R Native’s Facebook page has over 50,000 followers, with 77% accessing via a mobile device. 

Today, nearly 20% of We R Native’s users access mental health content on the website, and almost one-third of its 400+ Ask Auntie questions address mental health questions or concerns. Given the widespread use of mobile phones by this population, and their interest in mental health topics, mobile health (mHealth) interventions provide a highly accessible way to reach AI/AN youth and young adults who live across vast geographies in urban and rural communities.

### 1.2. Efficacy of Technology-Based Interventions 

Technology-based interventions are increasingly being used to promote adolescent health [13]. Several studies have reported improvements in mental health outcomes [14,15,16,17]. Further, mHealth interventions (delivered to youth using text messaging and social media) have been successfully used to improve help-seeking skills [18], including suicide prevention skills [14,19]. 

In 2014, We R Native evaluated its text messaging service to determine whether culturally tailored text messages could improve youth’s sexual health knowledge, attitudes, and behavior towards condom use and /sexually transmitted infections/ human immunodeficiency virus (STI/HIV) testing [20]. Over 408 AI/AN teens and young adults enrolled in the study: Texting 4 Sexual Health. Two text messages were sent per week for 12 weeks. Pre- and post-surveys were completed via text message. Study results showed a positive change in both attitude and behavior towards condom use, and intention and behavior towards STI/HIV testing. The study confirmed that text messages are an effective and efficient way to deliver essential health information to AI/AN youth [20]. However, technology-based interventions like We R Native and Texting 4 Sexual Health are only useful if they can recruit, retain, and engage program participants. 

Few mHealth interventions have been designed for AI/AN teens and young adults. No studies that we are aware of have reported how to enroll and engage AI/AN teens and young adults primarily using virtual platforms. Previous research indicates there are no standards for how to recruit or retain participants in mHealth intervention trials [21]. 

We identified two recent examples of social-media-based outreach efforts designed specifically for Native participants. The first—Native Women Young Strong Empowered (WYSE) Choices—recruited urban AI/AN young women to participate in study interviews using Facebook and Twitter posts [22]. Led by the Colorado School of Public Health Native WYSE Choices research team, in partnership with Ingenious (app developers), the team is carrying out formative research to design and evaluate an mHealth App to reduce the risk for alcohol exposed pregnancy. A second recruitment effort—the Indigenous Futures Project (IFP)—sought participants for a survey designed to identify core issues that impact Indian Country and Alaska. Led by the Center for Native American Youth, IllumiNative, and the Native Organizers Alliance, the team used social media posts to recruit survey participants (http://indigenousfutures.illuminatives.org/). Information about the recruitment, enrollment, and engagement of participants using social media in these two examples have not yet been published. 

### 1.3. Goal of this Study

This paper will share lessons learned recruiting and enrolling participants via social media in the BRAVE study—a national, multiphase project to design and evaluate a text message and video-based behavioral intervention—and tips to support campaign engagement. 

### 1.4. Research Partners

The BRAVE study’s efficacy phase was carried out with collaboration between the NPAIHB’s Tribal Health—Reaching out InVolves Everyone (THRIVE) and We R Native projects and the mHealth Impact Lab. The NPAIHB is a tribal non-profit organization representing 43 federally recognized tribes in Washington, Oregon, and Idaho. The mission of the NPAIHB is to “eliminate health disparities and improve the quality of life of American Indians and Alaska Natives by supporting Northwest Tribes in their delivery of culturally appropriate, high-quality health care.” The NPAIHB’s governing board meets quarterly and is composed of one delegate from each member tribe, selected by the individual tribal governments. The Northwest Tribal Epidemiology Center (NW TEC) is housed under NPAIHB. It provides support in the way of research, surveillance, and public health capacity building in partnership with the 43 federally recognized Northwest region tribes. 

Housed in the Colorado School of Public Health, the mHealth Impact Lab was founded in 2015, with a mission to facilitate the rapid and rigorous development, implementation, and evaluation of mobile and digital technology for health promotion and disease prevention that address inequalities in health outcomes. They work with researchers to develop and test innovative technology-based solutions using text messaging, social media, apps and sensors to improve health outcomes and clinical care; they partner with industry to create the evidence base for mobile and digital solutions intended to impact health outcomes; and they generate new knowledge on approaches to using emerging technologies to facilitate self-management of illness and promotion of wellness among diverse populations. 

## 2. Materials and Methods 

### 2.1. Overview

All data collection methods were approved by the Portland Area Indian Health Service Institutional Review Board (PA IHS IRB) in Portland, OR (PI: Craig Rushing, Protocol #: 1384639). All instruments were reviewed and approved by the IRB before data collection took place.

### 2.2. Communication Theory

To develop messages that resonate with AI/AN teens and young adults, the NPAIHB uses “social marketing,” an evidence-based planning process that has been shown to improve the impact of health messages [23]. The framework uses iterative phases of review to ensure campaign products reflect the target audience’s preferred language, tone, attitudes, norms and values. Learning and health communication theories support culturally tailored media to increase behavior change [24,25,26]. 

Tailored information is more likely to be read, understood, perceived as personally relevant, and remembered [27]. Rushing and Stephens have developed guidelines for designing culturally relevant technology-based health interventions, with the encouragement to include medically accurate age- and gender-appropriate content, be holistic and real (“reflect the unique life experiences of Native youth and address the root social determinants of their health”), be based in culture, focus on assets and skills, encourage dialogue with trusted adults, be interactive, include evaluation plans to monitor use and assess impact [28]. This guidance informed the development of the BRAVE intervention and STEM control messages. 

### 2.3. Behavioral Intervention

Teens and young adults enrolled in this study received either: 8 weeks of BRAVE text messages or 8 weeks of STEM text messages; and then crossed over to the other arm and received the next set of messages (see Appendix A).

#### 2.3.1. BRAVE Campaign Messages

The BRAVE campaign included 3–5 text messages per week, including 1 role model video per week and a related image. The intervention was designed to amplify and reinforce healthy social norms and cultural values, teach suicide warning signs, prepare youth to initiate difficult conversations with peers and trusted adults, encourage youth to access mental health resources (i.e., tribal clinics, chat lines), destigmatize mental health services and connect youth to trusted adults. 

#### 2.3.2. BRAVE Role Model Videos

The message series included links to 7 role model videos (1–3 min each) that featured relatable characters experiencing and addressing violent behavior, alcohol misuse, and suicidality (through the eyes of a perpetrator, an IPV survivor, and a peer bystander), intended to demonstrated important coping and help-seeking skills. 

#### 2.3.3. STEM Campaign Messages

The STEM campaign included 3–5 text messages per week, including 1 role model video per week and a related image. Participants in the STEM arm received messages designed to elevate and re-affirm Native voices in science, technology, engineering, math and medicine (STEM), as shown in Table 1. 

### 2.4. Eligibility Criteria

This study included self-identified American Indian and Alaska Native youth aged 15–24 years old. All participants were required to have a cell phone with text message capabilities. Eligibility was described via text message outlined below and confirmed using the pre-survey.
We R Native is doing a study to evaluate a text messaging program and we think you might be eligible. Earn up to $40 for your time. It takes courage to change or to step up and help a friend. Reply BRAVE if you’re interested.

Those who were not AI/AN or 15–24 years old received a text message, noting their ineligibility:
We R Native: Thanks for your interest in our study, but you must be AI/AN and 15–25 years old to participate. If you would like to receive text messages about science, engineering, technology, or medicine, reply with “STEM.” Or check out our other health resources at www.weRnative.org


### 2.5. Study Recruitment

From September to December 2019, we recruited AI/AN teens and young adults via We R Native’s social media channels (Facebook, text message, Instagram). Additional recruitment took place through listservs associated with tribes, tribal health organizations, Indian education and human service organizations that serve AI/AN teens and young adults (i.e., Indian Health Service, Meth Suicide Prevention Initiative, Bureau of Indian Education, Healthy Native Youth, and Tribal Colleges). 

Interested youth were asked to text the keyword BRAVE to the shortcode 97779, which triggered a series of eligibility and consent text messages, including a link to an online consent form for more information about this study.
Super! If you participate, you will receive either: 8 weeks of wellness messages designed to promote cultural pride and resilience, or 8 weeks of STEM messages designed to elevate Native voices in science, technology, engineering, and math.
Afterward, the two groups will switch and you’ll receive the other set of messages. Altogether, you’ll get 3–5 health msgs per week for 4 months. Earn $10 for answering four sets of Q’s over 8 months. More info about study is available at: http://lil.ms/1v5w

You may leave the study at any time by texting STOP. We will send you a pre-survey when the study begins. 

### 2.6. Enrollment

To enroll in this study, participants were required to complete the pre-survey. Those who met the eligibility requirements and completed the pre-survey were randomized into this study (*n* = 1030). Those who expressed interest received up to three reminders to complete the pre-survey. Participants received a $10 amazon gift code for each survey they completed, up to $40 per person in appreciation for their time.

### 2.7. Brave Study Ad Design and Placement

Study recruitment Ads were placed on Facebook and Instagram and were managed using Facebook Ads Manager. Each platform had unique specifications for Ad design, target audience, and dates of deployment. Facebook Ads cannot be targeted to AI/AN teens and young adults, so user “interests” were used for Ad targeting, including Location: United States; Age: 15–24; People who match interests: National Museum of the American Indian, National Indian Education Association, Institute of American Indian Arts, Navajo Times, Indian Country Today Media Network, American Indian College Fund, Powwow, Indigenous Rights, Native American Journalist Association, National Congress of American Indians, Indigenous Peoples, A Tribe Called Red, Shoni Schimmel or Native News Online. Ads with a positive tone typically received greater reach and impressions than posts simply describing study eligibility criteria, Table 2. 

### 2.8. Social Media Ads, Examples, Conversion Rates and Costs

The team used data from Facebook and Instagram to explore participant retention and message engagement. Measures included the following: (1) impressions, which describe the number of Ads displayed and if the ad was clicked; (2) link clicks, which is the number of times a participant clicked on a link to view a linked website; (3) post engagement, which is the number of Ads a participant clicked on; and (4) ThruPlay, which is an ad optimization platform for video campaigns. Altogether, we spent over $5000 on recruitment Ads, reaching over 567,000 social media users. Examples of social media recruitment platforms, link clicks, rates, and costs are summarized in Table 3. 

### 2.9. Analysis

Our team monitored inconsistencies in pre-survey data to confirm eligibility and ensure participants were not enrolled in this study more than once. The study team also exported analytics from Facebook and Instagram to assess patterns in recruitment regularly. Recruitment data are based on text message opt-in dates, and study completion rates were determined based on the number of eligible participants randomized into this study and the number of eligible participants that completed the BRAVE and STEM arm.

## 3. Results

### 3.1. SMS Message Analytics

Across the two study arms, we sent 164,539 text messages and received 5780 responses from 1030 subscribers. Total messages sent includes the sum of each individual text message sent to subscribers’ cell phones (and the count of all parts, if any were sent as multipart SMS) in both arms of this study. The number of text messages sent to each individual varied based on the participants level of engagement, which includes messages received from subscribers. For example, if a participant replied to a message, they would receive an automated response, increasing both the number of messages sent and message engagement, as shown in Figure 1.

#### 3.1.1. Study Retention

Participant retention was high across the two study arms. For BRAVE, 41 participants opted out during the intervention, and 25 opted out at crossover. For STEM, 45 participants opted out during the control, and 18 opted out at crossover. In total, 86 participants opted out of this study during the first arm, and 43 opted out after the crossover, resulting in a 87% retention rate.

#### 3.1.2. Message Engagement

The SMS messages included several strategies to track and encourage participant engagement with the intervention as a skill building tool. Participants could click video links, text MORE (when prompted) to learn more about important topics and were given periodic “calls to action” that prompted self-reflection, goal setting, or were encouraged to practice coping and help-seeking skills. Examples of each strategy include:
Video links requiring user engagement: Hey it’s Alex. I didn’t see it at the time, but Chris was my everything. The pow to my wow. The fry to my bread. The first real love of my life besides bball, and I f***ed it all up: Episode 2 (https://www.youtube.com/watch?v=9aRxIQd_62E&feature=youtu.be)
Text MORE engagement: Alex’s drinking was out of control. Drinking in moderation is defined as having no more than 1 drink per day for women and no more than 2 drinks per day for men. Drinking too much can cause lots of negative outcomes. Text MORE to learn more. 
Additional MORE message: Regular excessive alcohol use is associated with hangovers, reduced sexual performance, aggressive behavior, accidents and injury, anxiety and depression, relationship difficulties, and suicide... just to name a few. Drinking in moderation (or not drinking at all) is the best way to keep in control.
Skill practice engagement: Many of us have grown up in traumatic environments. Now it’s up to us to break the cycle. Having people in our lives who love and care about us is one way to build resilience. Week 2 Challenge: Reach out to a friend or family member this week. Let them know you appreciate them and their support.

Several patterns emerged in user engagement, across the two study arms. Participant engagement and interaction with the study messages dropped off over time. Links to external sources of information were accessed by viewers, but the frequency of links clicked stayed higher when fewer links were offered. At 20 messages with light links (1 link per 3 SMSs), the interaction rate was greater than 4%. In sequences with more frequent links (2 links per 3 SMSs), the interaction rate was less than 4%. The greatest click engagement was with links to the BRAVE role model videos, compared to other external resources, as shown in Table 4.

#### 3.1.3. Message Engagement with a Call to Action

Text messages that included “a call to action” received greater response rates than those that did not. Messages promoting culture had an interaction rate that was two times greater than plain text messages. Engagement with messages prompting users to request MORE remained high, even later in the campaigns when respondent fatigue was evident. Reply rates stayed relatively high in comparison with links and plain text. When users follow up with MORE or a REPLY, they were often higher interaction users. Post click rates on secondary messages averaged over 25% when looking at both campaigns together. 

## 4. Discussion

This paper summarizes the process of recruiting and engaging AI/AN teens and young adults in a culturally relevant behavioral health intervention in the U.S. We believe mHealth interventions can amplify and reinforce healthy social norms and cultural values, teach suicide warning signs, prepare youth to initiate difficult conversations with peers and trusted adults, encourage youth to access mental health resources (i.e., tribal clinics, chat lines), destigmatize mental health, and connect youth to trusted adults. Research testing the effectiveness of text-based interventions has yet to focus on AI/AN teen and young adults’ mental health outcomes. As a result, there are no rigorously evaluated interventions using these ubiquitous messaging platforms and technologies. Principal findings learned from BRAVE about recruitment and enrollment may guide the mHealth interventions for AI/AN teens and young adults.

### 4.1. Principal Findings

Recruitment and enrollment in the BRAVE study was supported by the We R Native multimedia health resources at the NPAIHB. With more than 1 million social media views per year, recognized as a trusted health resource, it was possible to recruit and enroll AI/AN teens and young adults in the BRAVE study. Living in urban and rural communities across the U.S., AI/ANs are generally considered a hard-to-reach population. Bridging theses vast territories, mHealth interventions make it uniquely possible to recruit and retain a national sample of AI/AN teens and young adults in behavioral health studies. NPAIHB utilized multiple recruitment methods to build community interest in BRAVE, including e-Newsletters, listservs, conferences, print postcards, tribal epidemiology centers, websites, social media native influencers, and Native student centers at schools and universities. During the recruitment phase, Facebook Ads with a positive tone received more impressions than Ads that simply described this study. This tells us that tone and content are essential to recruitment strategies in mHealth interventions for this population. 

Incentivizing participation with a $10 Amazon gift card for each survey completed contributed to the high level of retention across both study arms. Reasons for participant opt out may be related to text message fatigue, loss of access to technology that would allow text interaction, or low levels of satisfaction with content provided. Building in additional incentives for user engagement may reduce the number of opt outs in future mHealth interventions (see Appendix A). 

Lack of response was evident in both groups after the 10th message. This finding is consistent with previous studies where responses decreased over time, suggesting text message fatigue [29]. Building in a ‘pause’ or a break around the 12th text message might be a promising strategy for future campaigns. Doing so might help maintain a higher level of participant engagement that was observed at the beginning of the campaign. Future studies should explore the optimal number and frequency of messages to enhance retention and behavior change. 

Not surprisingly, text messages that included “a call to action” received greater response rates than those that did not. With regard to content, BRAVE text messages with calls to action need to have text interactivity, instead of asking users to do something and then reply via text on how it went. This is consistent with previous studies in other populations, where bi-directional text messaging and dynamic messaging resulted in greater participant engagement [30]. 

### 4.2. Limitations

The phase of the research had several strengths and limitations. We used convenience sampling methods, and therefore, the findings are not representative of the entire population of the AI/AN teen and young adult population. The sample was mainly recruited through We R Native, which is a multimedia health and wellness resource. Their participation in We R Native may have influenced how they viewed the BRAVE study and their participation in it. Additionally, this study utilized Facebook and Instagram social media platforms. The use of more popular platforms such as Instagram, Snapchat, and TikTok may have resulted in different recruitment strategies, retention, and content. Finally, external factors such as schoolwork, weather, holidays, and access to social media likely contributed to enrollment and retention issues, but these factors were not evaluated in this study.

### 4.3. Strengths

Our study builds seamlessly upon the digital research activities already carried out by the Northwest Portland Area Indian Health Board and the mHealth Impact Lab and will fill a critical need for evidence-based interventions that reflect the unique needs and worldview of AI/AN teens and young adults. Most importantly, We R Native is a trusted resource for Native teens and young adults, with well-established communication channels. The length of the project, the commitment and leadership of NPAIHB, the diversity of AI/AN teens and young adults involved, and the support from content experts provided consistent support for the BRAVE study. BRAVE’s focus on culturally relevant images, language, and peers as role models demonstrated respect for diversity, values, and translation of health communication messaging that AI/ANs could identify with. Lessons about recruitment and retention outlined in this paper contribute to the literature by describing how to enroll and engage AI/AN teens and young adults utilizing multiple social media platforms at a relatively low cost. 

## 5. Conclusions

Teens and young adults utilize cell phones and text messaging more than other groups [30]—mHealth interventions are increasingly being used to promote health, build help-seeking skills, and prevent suicide. Findings presented in this study demonstrate possibility and promise for culturally tailored mHealth interventions that actively engage AI/AN teen and young adult populations. 

Future papers will examine to what extent BRAVE intervention messages improve youth’s resilience, self-esteem, self-efficacy, social support, and their use of mental health resources. Research will also explore if higher levels of engagement result in a greater effect. If deemed effective, the BRAVE intervention has the potential to improve the relevance, efficacy, and utilization of mental health resources delivered through We R Native’s messaging channels—reaching a high-risk, underserved population—and will create new mechanisms to monitor and evaluate the impact of mHealth interventions. Resultant data collection tools and processes can inform our collective evidence surrounding technology and teen and young adult mental health and offer guidance for promoting their adoption and use within and beyond Indian Country.

## Figures and Tables

**Figure 1 ijerph-17-09437-f001:**
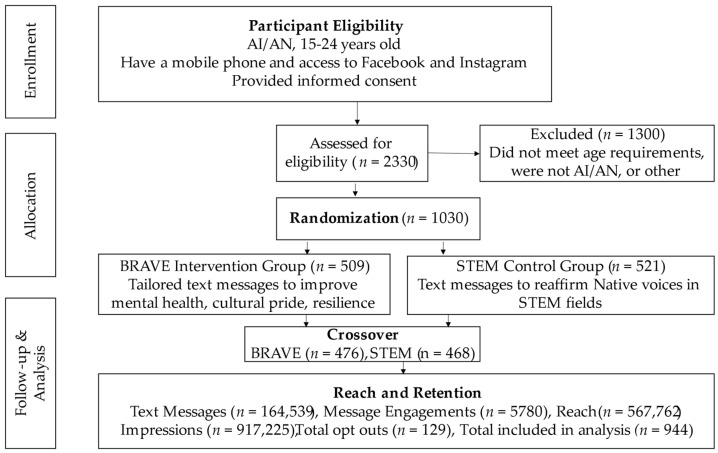
This is a figure outlining participant recruitment, randomization, reach, and retention.

**Table 1 ijerph-17-09437-t001:** Example of BRAVE and STEM arm messages.

BRAVE Arm Messages—Week 2	STEM Arm Messages—Week 2
Sunday Hi! I’m Alex. I’ve been through a lot this year—parties, girl drama, rez life. To really get to know me, you should see where I grew up. Watch this video to learn more about me: Alex—Episode 1 (https://www.youtube.com/watch?v=pPBI8oUXZ8Q) Follow-up Text: As you can see, my dad taught me everything I know—the good and the bad. My dad’s drinking, the arguments that would follow… I thought those things were normal.	Sunday [Emojis—OK, Sunglasses, Pointing Left; $ signs Face] Cool? Wanna make some $? The Bureau of Labor predicts a 27% growth in IT jobs by 2024... Web development and computer science are ideal jobs in rural (and urban) communities. People who can do the work are highly sought after and well paid, and you can often work remotely. You can even use coding skills to help preserve Native languages: https://www.youtube.com/watch?v=pPBI8oUXZ8Q Follow-up Text: We’re descended from scientists, inventors, and innovators. Imagine bridging the digital divide in native communities!? Learn tech to continue our traditions.

**Table 2 ijerph-17-09437-t002:** Social media recruitment ad name summary and impressions.

Ad Name	Impressions
FB Post: “We Need Your Help! We R Native is recruiting…”	12,982
FB Post: “🤳 Text BRAVE to 97779”	12,766
Instagram Post: An Auntie’s hug and laugh…	114,713
Instagram Post: Are you a Native teen or young…	124,492
Instagram Post: How can we help our friends who…	68,876

**Table 3 ijerph-17-09437-t003:** BRAVE social media text examples, impressions, link clicks, conversion rates, and costs.

Platform/Ad Type	Text Example	Impressions	Link Clicks	Conversion Rate (%)	Ad Costs (US$)	Cost per Link Click (US$)
Instagram Link Clicks	I am building a better world for…	550,880	353,150	0.64	$2606	$2.69
Facebook Post Link Clicks	We Need Your Help! We R Native is recruiting…	25,748	12,020	0.46	$119	$0.65
Facebook Post Engagement	I am building a better world for our people…	188,550	111,081	0.58	$1906	$0.12
Facebook ThruPlay	Abuse isn’t always physical. Abuse can come in…	152,047	91,511	0.60	$510	$0.05
Total		917,225	567,762		$5141	

**Table 4 ijerph-17-09437-t004:** Link clicks by message number and total.

Message #	Original Link	Clicks (Unique/Total)
3	https://www.wernative.org/articles/how-does-a-person-become-resilient	97/132
4	https://www.youtube.com/watch?v=dOcthWY9CLI&feature=youtu.be&ab_channel=weRnative	128/168
18	https://www.youtube.com/watch?v=9aRxIQd_62E&feature=youtu.be	113/150
20	https://www.strongheartshelpline.org/abuse/	46/66
22	https://www.loveisrespect.org/ https://www.loveisrespect.org/	28/36
25	https://www.youtube.com/watch?v=k3YycAzDYkc&feature=youtu.be&ab_channel=weRnative	60/73
29	https://www.wernative.org/articles/creating-safe-spaces	29/48
32	https://www.youtube.com/watch?v=SfaJBE7ttdk&feature=youtu.be&ab_channel=weRnative	50/79
36	https://www.facebook.com/TraditionNotAddiction	17/25
39	https://www.youtube.com/watch?v=VpW035EXdQM&feature=youtu.be	42/70
41	https://www.strongheartshelpline.org/	16/27
46	https://www.youtube.com/watch?v=f8EubVYcrB0&feature=youtu.be	37/48

# Refers to the SMS message sent to participants.

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
