# Peer review of "Recruiting and Engaging American Indian and Alaska Native Teens and Young Adults in a SMS Help-Seeking Intervention: Lessons Learned from the BRAVE Study"

_ijerph, 2020, doi:10.3390/ijerph17249437_

Round 1

Reviewer 1 Report

This paper shares lessons learned recruiting and engaging participants in the BRAVE study. Lessons learned from this process may help teen and young adult-serving organizations, prevention programs, policymakers, researchers, and educators as they support the next generation of AI/AN change-makers. The followings are the comments on some key issues.

  1. Because they do not review relevant literature or address a well-developed theory, their findings will lose its value for academia.

  2. The paper has a review of related literature, however, some relevant literature should be enhanced.

  3. The results should be presented more accurately. There is a disconnection between discussion and results.

Author Response

This paper shares lessons learned recruiting and engaging participants in the BRAVE study. Lessons learned from this process may help teen and young adult-serving organizations, prevention programs, policymakers, researchers, and educators as they support the next generation of AI/AN change-makers.

Thank you.

Because they do not review relevant literature or address a well-developed theory, their findings will lose its value for academia.

We appreciate this comment. We reviewed relevant published literature based on what is available. We do not address a well-developed theory because that was not the goal of this paper. The goal of this paper was to describe enrollment and engagement of AI/ANs in an mHealth intervention.

The paper has a review of related literature, however, some relevant literature should be enhanced.

We agree and added literature throughout, thank you. Please see the updated reference list, see lines 76-82 and 110-119.

The results should be presented more accurately. There is a disconnection between discussion and results.

We appreciate this suggestion. We moved the message engagement section to the methods section based on a previous reviewer comment. Since the results of this study are about the reach of SMS messages, retention, and engagement—it may feel a bit different than a traditional research article that is presenting findings from a study. As we mentioned in the manuscript, findings from the BRAVE RCT are being published in a separate paper, our goal with this manuscript is to share valuable information about recruitment and engagement with this population. If you have a specific comment about the disconnect we would be happy to address that. Thank you.

Reviewer 2 Report

This paper was very well-written and is much needed in the literature as others increasingly seek to do similar work - i.e. using mobile health technology and social media to reach Native populations across the country for intervention research.  The only suggestion I have is that the paper be reviewed carefully to ensure there are no lingering grammatical or punctuation errors or missing words, and that all statements are properly referenced.  There is also a correction to the Native WYSE study name...it should read Native Women Young, Strong, and Empowered making CHOICES (all caps as "CHOICES" is an acronym for the intervention on which the study intervention is based).

Author Response

This paper was very well-written and is much needed in the literature as others increasingly seek to do similar work - i.e. using mobile health technology and social media to reach Native populations across the country for intervention research.  The only suggestion I have is that the paper be reviewed carefully to ensure there are no lingering grammatical or punctuation errors or missing words, and that all statements are properly referenced.  There is also a correction to the Native WYSE study name...it should read Native Women Young, Strong, and Empowered making CHOICES (all caps as "CHOICES" is an acronym for the intervention on which the study intervention is based)

Thank you for your thoughtful review. We conducted reviewed the manuscript for grammatical and punctuation errors. These changes can be found throughout the document using the MS Word tracking feature.

We revised this mention on lines 117 and 118. Thank you.

Reviewer 3 Report

I think this a very interesting article and a very interesting research that I really liked to read. I think this work has a lot merit but also hast some aspects that perhaps could be revised to improve the overall quality of the article:

  1. There are abbreviations in the abstract (AI/AN) that should be explained before using them.
  2. Eligibility Criteria. In these studies the potential selection bias is very important. For example, economic incentives can generate an important selection bias and it can induce some participants to practice gaming to obtain the incentives or even more incentives that they are supposed to obtain. So, this aspect could be more detailed and also described in limitations section. The authors could explain how they have managed the potential gamification practice of users.
  3. In Study Recruitment the authors describe that «From September to December 2019, we recruited AI/AN teens and young adults via We R Native’s social media channels (Facebook, text message, Instagram)». In Brave Study Ad Design and Placement the authors write that «Study recruitment Ads were placed on Facebook and Instagram and were managed using Facebook Ads Manager». These two aspects can generate a considerable selection bias because using social media excludes potential participants who do not use social media or are not very active or were not active during the selection window. And there is another potential bias generated for those who participated, because of the auto selection bias. The authors could explain how they have managed this possible selection bias
  4. In Social Media Ads, Examples, Conversion Rates and Costs, the authors write that «we spent over $5,000 on recruitment Ads, reaching over 567,000 social media users». It is easy to understand the use of paid Ads. The problem of using paid Ads is that the designers of the study had no control on how these Ads are showed, as only the owners of the different social media channels have full control. As a consequence, it is difficult to know if the study can be repeated in similar conditions and this could affect to its external validity. Of course this does not invalidate the study, but the authors perhaps could detail better these aspects.
  5. In Study retention, the authors describe a 87% retention rate. It seems adequate, but we must consider that there is an economic incentive that can be influencing this retention rate. So, this aspect could be described.
  6. The Message Engagement section, included in Results, perhaps could fit better in Method section. 
  7. In the Discussion section, there are only two references. This section should be used to compare and discuss their findings with those from other authors. I understand that sometimes is very difficult to find previous research to compare findings, but I think that this section would be better if the authors were able to add more references, if possible.
  8. In the limitations sections, perhaps the authors could detail better some important aspects: auto selection bias, potential bias of economic incentives, or the limitations of using social media and paid Ads for the recruitment.

Overall, I think that this is a very good and interesting research. It can benefit from some small revisions, like the ones described, but I think the authors have performed a great work. Congratulations.

Author Response

I think this a very interesting article and a very interesting research that I really liked to read. I think this work has a lot merit but also hast some aspects that perhaps could be revised to improve the overall quality of the article:

Thank you.

There are abbreviations in the abstract (AI/AN) that should be explained before using them.

This has been revised and corrected on line 23 and 24.

Eligibility Criteria. In these studies the potential selection bias is very important. For example, economic incentives can generate an important selection bias and it can induce some participants to practice gaming to obtain the incentives or even more incentives that they are supposed to obtain. So, this aspect could be more detailed and also described in limitations section. The authors could explain how they have managed the potential gamification practice of users.

We considered selection bias and economic incentives and added a sentence about economic incentives on line 418.

Interventions like BRAVE are particularly needed in the current pandemic where in-person activities and support are limited.

While the intervention does invite engagement from participants, it is not a game in which users can self-select content to view or receive different “doses” based on usage – all participants received the same set of text messages, with no added incentive to “engage” with that content. Therefore we feel that aspects of gamification practice do not apply to BRAVE study participants.

It should be noted that there is sufficient evidence in the literature that supports gamification as an approach to support health and wellbeing, Johnson, D; Deterding, S; Kuhn, KA; Staneva, A; Stoyanov, S; Hides, L. Gamification for health and wellbeing: A systematic review of the literature. Internet Interv. 2016, 6:89-106. https:// doi:10.1016/j.invent.2016.10.002

In Study Recruitment the authors describe that «From September to December 2019, we recruited AI/AN teens and young adults via We R Native’s social media channels (Facebook, text message, Instagram)». In Brave Study Ad Design and Placement the authors write that «Study recruitment Ads were placed on Facebook and Instagram and were managed using Facebook Ads Manager». These two aspects can generate a considerable selection bias because using social media excludes potential participants who do not use social media or are not very active or were not active during the selection window. And there is another potential bias generated for those who participated, because of the auto selection bias. The authors could explain how they have managed this possible selection bias.

We appreciate this comment and addressed in on lines 416 to 423. We feel that our recruitment methods (social media, primary approach; asking educators to enroll students, secondary approach) definitely had biases, but we felt those limitations aligned with the prerequisites for participating in the intervention (access to a cell phone with reliable connectivity), and was thus an informed tradeoff.

This study in no way represents the entire AI/AN teen and young adult population, so these limitations and the results should be considered within the context in which BRAVE was designed and implemented. Thank you.

In Social Media Ads, Examples, Conversion Rates and Costs, the authors write that «we spent over $5,000 on recruitment Ads, reaching over 567,000 social media users». It is easy to understand the use of paid Ads. The problem of using paid Ads is that the designers of the study had no control on how these Ads are showed, as only the owners of the different social media channels have full control. As a consequence, it is difficult to know if the study can be repeated in similar conditions and this could affect to its external validity. Of course this does not invalidate the study, but the authors perhaps could detail better these aspects.

We used the Ads to target our existing social media users (and their friends and family), which also biased the participants towards those who were already familiar with We R Native and our health messages. Likewise we felt this group, with pre-existing trust and familiarity in WRN, would be most likely to fully engage with the intervention.

In Study retention, the authors describe a 87% retention rate. It seems adequate, but we must consider that there is an economic incentive that can be influencing this retention rate. So, this aspect could be described.

We appreciate this comment. On line 353 of the original text we noted that incentivizing participants likely contributed to higher rates observed.

We added a statement on line 355 and also a citation to support the use of financial incentives to retain participants.  

The Message Engagement section, included in Results, perhaps could fit better in Method section.

We appreciate this comment. We moved lines 283 to 309 of the message engagement section to the materials and methods section beginning on line 260. We left our analysis and Table four in the result section, since these are results from this study.

In the Discussion section, there are only two references. This section should be used to compare and discuss their findings with those from other authors. I understand that sometimes is very difficult to find previous research to compare findings, but I think that this section would be better if the authors were able to add more references, if possible.

Thank you. We added additional references as appropriate.

In the limitations sections, perhaps the authors could detail better some important aspects: auto selection bias, potential bias of economic incentives, or the limitations of using social media and paid Ads for the recruitment.

We appreciate this comment and added more information about bias and limitations of using paid Ads for recruitment, see lines 443-460.

Overall, I think that this is a very good and interesting research. It can benefit from some small revisions, like the ones described, but I think the authors have performed a great work. Congratulations.

Thank you. We have made numerous revisions and appreciate your support.